# *ERCC1* and *ERCC2* Polymorphisms Predict the Efficacy and Toxicity of Platinum-Based Chemotherapy in Small Cell Lung Cancer

**DOI:** 10.3390/pharmaceutics16091121

**Published:** 2024-08-25

**Authors:** Andrés Barba, Laura López-Vilaró, Malena Ferre, Margarita Majem, Sergio Martinez-Recio, Olga Bell, María J. Arranz, Juliana Salazar, Ivana Sullivan

**Affiliations:** 1Department of Medical Oncology, Hospital de la Santa Creu i Sant Pau, 08041 Barcelona, Spain; abarba@santpau.cat (A.B.); mmajem@santpau.cat (M.M.); smartinezre@santpau.cat (S.M.-R.); 2Department of Medicine, Faculty of Medicine, Universitat Autònoma de Barcelona, 08035 Barcelona, Spain; 3Department of Pathology, Hospital de la Santa Creu i Sant Pau, 08041 Barcelona, Spain; llopezv@santpau.cat (L.L.-V.); mferref@santpau.cat (M.F.); 4Translational Medical Oncology Laboratory, Institut de Recerca Sant Pau (IR Sant Pau), 08041 Barcelona, Spain; obell@santpau.cat; 5Research Laboratory Unit, Fundació Docència i Recerca Mútua Terrassa, 08221 Terrassa, Spain; mjarranz@mutuaterrassa.es

**Keywords:** *ERCC1*, *ERCC2*, platinum-based chemotherapy, small cell lung cancer, pharmacogenomics

## Abstract

Standard first-line chemotherapy in small cell lung cancer (SCLC) is based on the platinum plus etoposide combination. Despite a high objective response rate, responses are not durable and chemotherapy-induced toxicity may compromise treatment. Genetic variants in genes involved in the DNA-repair pathways and in etoposide metabolization could predict treatment efficacy and safety and help personalize platinum-based chemotherapy. Germline polymorphisms in *XRCC1*, *ERCC1*, *ERCC2*, *ABCB1*, *ABCC3*, *UGT1A1* and *GSTP1* genes were investigated in 145 patients with SCLC. The tumor expression of ERCC1 was determined using immunohistochemistry, and the tumor expression of ERCC1-XPF was determined via a proximity ligation assay. Survival analyses showed a statistically significant association between the *ERCC1* rs11615 variant and median progression-free survival (PFS) in patients with limited-stage (LS) SCLC (multivariate: hazard ratio 3.25, [95% CI 1.38–7.70]; *p* = 0.007). Furthermore, we observed differences between the ERCC1-XPF complex and median PFS in LS-SCLC, although statistical significance was not reached (univariate: positive expression 10.8 [95% CI 4.09–17.55] months versus negative expression 13.3 [95% CI 7.32–19.31] months; *p* = 0.06). Safety analyses showed that the *ERCC2* rs1799793 variant was significantly associated with the risk of grade ≥ 3 thrombocytopenia in the total cohort (multivariate: odds ratio 3.15, [95% CI 1.08–9.17]; *p* = 0.04). Our results provide evidence that *ERCC1* and *ERCC2* variants may predict the efficacy and safety of platinum-based chemotherapy in SCLC patients. LS-SCLC patients may benefit most from ERCC1 determination, but prospective studies are needed.

## 1. Introduction

Small cell lung carcinoma (SCLC) is a poorly differentiated neuroendocrine tumor that accounts for approximately 15% of all lung cancers. It is characterized by rapid tumor growth and early development of metastasis [1]. The standard treatment for patients with limited-stage (LS) is the combination of platinum compounds plus etoposide (PE) chemotherapy and radiotherapy [2]. However, the disease progresses in half of the patients within 15 months [2], and no other treatment has yet shown to be superior to chemoradiotherapy. Patients with extensive-stage (ES) are treated with PE-based chemotherapy [3,4,5], and although 60% of patients initially respond, only 34% maintain this response at 6 months [4]. The combination of chemotherapy and immune checkpoint inhibitors (ICI) has become a new standard of treatment in ES-SCLC, but the benefit in overall survival (OS) is modest [3,4]. Furthermore, 60% of SCLC patients have severe toxicity to chemotherapy [3,4], and 3 to 19% of patients experience severe toxicity due to concurrent thoracic radiotherapy [2]. Advanced age and poor performance status (PS) have been correlated with higher toxicity and poorer survival [6,7]. Additionally, comorbidities and high pretreatment lactate dehydrogenase (LDH) levels have been correlated with an increased risk of mortality [8,9,10]. However, these factors do not accurately predict the clinical outcome in SCLC patients, and additional validated predictors of treatment efficacy and toxicity are needed.

Platinum compounds such as cisplatin and carboplatin are DNA alkylating agents. Their antineoplastic effects are decreased by the activation of cellular DNA repair mechanisms, mainly nucleotide excision repair (NER) and base excision repair (BER) pathways [11,12]. Ionizing radiation effects have also been shown to be affected by these DNA-repair pathways [13,14]. During DNA repair processes, excision repair cross-complementation group 1 (ERCC1) and xeroderma pigmentosum complementation group F (XPF) proteins form a heterodimer with endonuclease activity, which catalyzes the incision at the 5’-side of the damaged DNA strand [15]. In vitro studies have found that cell lines that are defective in terms of the ERCC1-XPF complex are more sensitive to cisplatin [16,17]. Low ERCC1 expression in tumor tissue has been correlated with better survival in non-small-cell lung cancer (NSCLC) [18,19,20] and in SCLC patients treated with platinum-based chemotherapy [21,22]. Furthermore, many studies have examined the influence of single-nucleotide polymorphisms (SNPs) in the *ERCC1* gene on clinical outcomes in platinum-containing regimens in NSCLC [23,24], but to a lesser extent in SCLC [25,26]. The most studied SNPs are *ERCC1* rs11615 located at codon 118 at exon 4, and *ERCC1* rs3212986 located in the 3′-untranslated region.

Etoposide is mainly metabolized through cytochrome P450 (CYP) 3A4 and 3A5, glutathione S-transferases (GSTs) and uridine 5′-diphosphate glucuronosyltransferase (UGT1A1). Genetic variants in the genes encoding these enzymes may contribute to the treatment toxicity in SCLC [27,28].

ERCC1 expression and variants in genes involved in DNA repair pathways and in etoposide pharmacokinetics may be predictors of efficacy and toxicity to PE-based treatment [18,22,24,27], but data concerning SCLC are insufficient and inconsistent [29,30,31]. Moreover, to the best of our knowledge, research exploring the role of ERCC1-XPF complex as a biomarker in SCLC is lacking. Our aim was to identify predictive biomarkers of clinical outcome in a cohort of SCLC patients treated with platinum plus etoposide chemotherapy. We analyzed polymorphisms in genes related to PE chemotherapy and radiotherapy in germline DNA, and evaluated the expression of ERCC1 and ERCC1-XPF proteins in tumor tissue.

## 2. Materials and Methods

### 2.1. Study Sample

We conducted a retrospective study in 145 patients with SCLC treated with PE-based chemotherapy between September 2007 and May 2021 at Hospital de la Santa Creu i Sant Pau (Barcelona, Spain). Blood samples were collected from all participants, and formalin-fixed paraffin-embedded (FFPE) tumor tissue samples were available from 106 (73%) of the 145. Table 1 summarizes the characteristics of the patients. This study was approved by the ethics committee of the Institut de Recerca Sant Pau (IIBSP-CCP-2022-56) and was performed in accordance with the Declaration of Helsinki. All patients gave informed consent for inclusion in this study.

### 2.2. Clinical Assessment

Progression-free survival (PFS) was defined as the time from initiation of PE chemotherapy treatment to disease progression according to Response Evaluation Criteria in Solid Tumors version 1.1 (RECIST 1.1) or death from any cause, whichever occurred first. OS was defined as the time from initiation of PE chemotherapy treatment to death from any cause. The objective response rate (ORR) was defined as the percentage of patients who achieved a partial or complete response to PE chemotherapy according to RECIST criteria 1.1 [32]. The PE-based chemotherapy-related toxicities analyzed were anemia, neutropenia and thrombocytopenia. Radiotherapy-related toxicities analyzed were esophagitis and pneumonitis. These toxicities were graded according to the Common Terminology Criteria for Adverse Events (CTCAE) Version 4.0. [33]. Hematological toxicities were dichotomized into grades 1–2 versus grades 3–4. Esophagitis and pneumonitis toxicities were dichotomized into their presence or absence.

### 2.3. SNPs Selection and Genotyping

We selected SNPs in genes involved in the DNA repair pathways (NER and BER) and etoposide pathway that have been previously associated with clinical outcomes in lung cancer patients treated with PE-based chemotherapy and/or radiotherapy [13,23,26,27,28,34,35,36,37] (see Table 2).

Genomic DNA from peripheral blood samples was obtained by automatic extraction (Autopure, Qiagen, Hilden, Germany). DNA concentration and quality were assessed using the NanoDrop 2000 spectrophotometer (Thermo Fisher Scientific, Wilmington, DE, USA). Genotyping was carried out by real-time PCR using TaqMan^®^ SNP genotyping assays on a 7900 HT Real Time PCR System (Applied Biosystems, Foster City, CA, USA). All methods were performed following the manufacturers’ recommendations. The *UGT1A1* rs3064744 (*UGT1A1*28*) variant was assessed as previously reported by Marcuello et al. [38]. The genotyping success rate was higher than 99% for samples and SNPs. The allele frequencies of the SNPs were consistent with those reported in European populations [39]. All SNPs were in the Hardy–Weinberg equilibrium.

### 2.4. Immunohistochemical Studies

ERCC1 protein expression was determined using immunohistochemistry (IHC) in FFPE tissues. Immunohistochemical stain was performed fully automated with Dako OMNIS Technology (Agilent Technologies, Santa Clara, CA, USA). We cut 4 µm thick FFPE tissue sections. Deparaffinized and rehydrated sections were heated for 30 min at 97 °C to perform antigen retrieval. These tissue sections were then stained with the ready-to-use anti-human ERCC1 mouse monoclonal antibody (clone 4F9, Dako Ltd., Cheshire, UK) followed by incubation with a secondary antibody and horseradish peroxidase coupled to a dextran polymer backbone. The slides were then revealed with 3,3′-diaminobenzidine tetrahydrochloride chromogen, counterstained with hematoxylin and coverslipped. The positive external control was tonsil. Internal positive controls were respiratory epithelium and/or lymphocytes present in a variable proportion in the sample cases. Two investigators (LL and MF) independently examined the nuclear ERCC1 staining and reviewed the discordant cases for mutual consensus. The ERCC1 expression pattern was classified as positive when nuclear staining was present, independently of intensity or proportion, and negative when nuclear staining was absent.

### 2.5. Proximity Ligation Assay (PLA)

We included samples with ERCC1-positive expression determined using IHC in the evaluation of nuclear colocalization of ERCC1 and XPF (n = 72). Samples with ERCC1-negative expression were considered negative for the ERCC1-XPF complex. We incubated 4 µm thick sections of FFPE tissues at 37 °C for 30 min and at 60 °C for 60 min with the primary antibodies anti-human ERCC1 mouse monoclonal antibody ready-to-use (clone 4F9, Dako Ltd., Cheshire, UK) and anti-XPF diluted at 1:200 (Polyclonal, Novus Biologicals, Centennial, CO, USA). The slices were then incubated with the anti-rabbit MINUS and anti-mouse PLUS Duolink^®^ PLA Probes (Sigma-Aldrich, St. Louis, MO, USA). The resulting oligonucleotides were hybridized, ligated, amplified and detected using the Duolink^®^ Detection Reagents for Brightfield (Sigma-Aldrich, St. Louis, MO, USA) according to the manufacturer’s instructions. Next, chromogenic PLA signals were visualized using optical microscope at 400× magnification. The presence of dark red colocalization signals (dots) in cell nuclei was considered positive for the ERCC1-XPF complex. Prior to the analysis, two investigators (LL and MF) observed the different expression patterns in 20 random samples to reach a consensus on the sample analysis. Four expression pattern groups were defined based on the number of signals within the cell nucleus (0, 1, 2, ≥3). The two investigators independently examined each full tissue section and selected at least three different representative areas to count the PLA signals. At least 50 tumor cells per area were analyzed. PLA signals were counted by establishing the percentage of nuclei in each of the formerly defined groups. Discordant cases were reviewed for mutual consensus. A semi-quantitative histochemical score (H-score) was calculated by multiplying the number of nuclear signals and the percentage of cells with nuclear signals. The H-score was considered positive for values ≥ 1 due to the low expression of the ERCC1-XPF complex (85% of tumors with H-score ≤ 25). Representative images were taken using scanned images from an Olympus Slideview VS200 and processed using the Olympus VS200 ASW 3.3 software (Olympus, Tokyo, Japan).

### 2.6. Statistics

Survival biomarkers were investigated in patients who received at least two cycles of PE chemotherapy and did not receive treatment with ICI (n = 121). The progression date was not available for four patients, and they were excluded from the PFS analyses (n = 117). ORR to chemotherapy biomarkers were studied in patients who received PE chemotherapy, excluding those who underwent ICI and concomitant chemo-radiotherapy (n = 99). PE-based chemotherapy-related toxicity analyses were conducted in all patients (n = 145). Radiotherapy-related toxicity analyses were conducted in patients who underwent thoracic radiotherapy (n = 40). The sample size had an 80% statistical power with a two-sided 95% CI to detect genetic effect sizes of moderate magnitude (OR ≤ 3), considering 90% mortality (Epi Info 7TM, https://www.cdc.gov/epiinfo; accessed on 15 May 2022). Kaplan-Meier estimates, log-rank tests and Cox regressions (with disease stage and eastern cooperative oncology group (ECOG) PS as covariates when appropriate) were used to study the influence of SNPs and proteins on survival analyses. Statistical differences between categorical variables were calculated using Chi-square or Fisher’s exact test based on sample size. Logistic regression analyses were performed considering toxicity as the dependent variable and polymorphisms and the appropriate clinical variables (disease stage, ECOG PS and platinum dose reduction) as predictors. Co-dominant, dominant and recessive inheritance models of genetic variants were used whenever appropriate. Statistical significance was set at less than 0.05. Correction for multiple comparisons was performed using the Bonferroni method (*p* < 0.0005). All statistical analyses were performed using IBM SPSS Statistics (version 26.0).

## 3. Results

### 3.1. Clinical Results

The median follow-up was 10.9 (interquartile range 6.5–22.3) months. In total, 107 patients (88.4%) showed disease progression, and 113 (93.4%) died. The median OS was 12.4 [95% CI 9.1–15.7] months, and the median PFS was 7.0 [95% CI 6.2–7.7] months. OS differed significantly according to the SCLC stage and ECOG PS. Patients with LS-SCLC had a median OS of 25.8 [95% CI 12.43–39.11] months compared to 9.7 [95% CI 7.83–11.52] months in patients with ES-SCLC (*p* < 0.001). Patients with ECOG PS 0–1 had a median OS of 15.5 [95% CI 12.45–18.63] months compared to 8.2 [95% CI 7.39–8.94] months in patients with ECOG PS ≥ 2 (*p* < 0.001). ES-SCLC patients with an ECOG PS of 0–1 (11.5 [95% CI 6.93–16.15] months) survived longer than patients with ECOG PS ≥ 2 (8.2 [95% CI 7.64–8.69] months) (*p* = 0.001). There were also significant differences in PFS according to the SCLC stage: 12.5 [95% CI 8.29–16.62] months in LS-SCLC compared to 6.1 [95% CI 5.33–6.80] months in ES-SCLC (*p* < 0.001).

### 3.2. Survival and Objective Response Rate Analyses 

#### 3.2.1. Analyses of Genetic Variants

Univariate analyses showed a statistically significant association between the *ERCC1* rs11615 variant and PFS (*p* = 0.02 in a dominant model) (Figure 1A). In the LS-SCLC subgroup, three SNPs showed statistically significant associations with PFS: *ERCC1* rs11615 (*p* = 0.003 in a dominant model) (Figure 1B), *ERCC2* rs50872 (*p* = 0.01 in a recessive model) and *UGT1A1* rs3064744 (*p* = 0.02 in a dominant model). In ES-SCLC patients, the *ERCC1* rs3212986 variant showed a statistically significant association with PFS (*p* = 0.049). Table 3 summarizes the statistically significant results. The full results are shown in Appendix A. Multivariate analyses showed the association between PFS and the *ERCC1* rs11615 variant in LS-SCLC patients (hazard ratio [HR] 3.25, 95% CI 1.38–7.70; *p* = 0.007). However, this association was not statistically significant after the Bonferroni test was applied.

Univariate analyses for OS showed a statistically significant association with the *ABCC3* rs4793665 variant (*p* = 0.047). Stratification by disease stage showed the *ABCC3* rs4793665 (*p* = 0.04) and the *ERCC1* rs11615 (*p* = 0.03 in a dominant model) variants were significantly associated with OS in LS-SCLC (Table 3). Multivariate analyses showed no statistically significant associations.

Analyses of ORR showed no significant correlations with the genetic variants investigated in any of the groups considered (total cohort, LS-SCLC and ES-SCLC subgroups) (Appendix A).

#### 3.2.2. Analyses of ERCC1 and ERCC1-XPF Complex Expression

In total, 11 of 106 FFPE tissues samples were not evaluable for ERCC1 protein expression due to insufficient representative tumor tissue. These samples were excluded from the analyses. No differences were observed in the 95 remaining patients regarding clinicopathological parameters in comparison with the 145 patients included in this study. A total of 72 samples were positive for ERCC1 expression, and 23 were negative (Figure 2A–C). Colocalization of ERCC1 and XPF proteins in the cell nuclei was positive in 65 samples (68.4%) (Figure 2D–F). ERCC1 expression in tissue samples did not correlate with the *ERCC1* rs11615 and rs3212986 polymorphisms.

Differences were observed between PFS and the expression of the ERCC1-XPF complex in the total cohort (n = 79), although these were not statistically significant. Patients with positive expression of the ERCC1-XPF complex achieved a shorter median PFS than patients with negative expression (6.5 [95% CI 5.68–7.30] months versus 10.2 [95% CI 4.19–16.20] months, respectively; *p* = 0.05). In the analyses according to disease stage, marginal differences were also observed between PFS and expression of the ERCC1-XPF complex in LS-SCLC patients (n = 26) (10.8 [95% CI 4.09–17.55] months versus 13.3 [95% CI 7.32–19.31] months; *p* = 0.06). No significant associations were observed between OS and ERCC1 or ERCC1-XPF complex expression.

### 3.3. Safety Analyses

#### Analyses of Genetic Variants

Univariate analyses for toxicity showed statistically significant associations between *ERCC2* rs50872 (*p* = 0.03 in a recessive model) and *XRCC1* rs25487 (*p* = 0.04 in a recessive model) and anemia. Moreover, *ERCC2* rs1799793 (*p* = 0.03 in a dominant model) and *UGT1A1* rs3064744 (*p* = 0.03 in a recessive model) were significantly associated with thrombocytopenia. Table 4 summarizes the statistically significant results. The full results are shown in Appendix A. In the multivariate analyses, the association between *ERCC2* rs1799793 and thrombocytopenia maintained statistical significance (odds ratio [OR] 3.15, 95% CI 1.08–9.17; *p* = 0.04). However, this association was not statistically significant after the Bonferroni test was applied.

Regarding radiotherapy-related toxicities in LS-SCLC, only five patients had pneumonitis, and the toxicity was not analyzed. No statistically significant associations were found for esophagitis (Appendix A).

## 4. Discussion

Chemoradiotherapy remains the treatment of choice for LS-SCLC, and the combination of immunotherapy and PE-based chemotherapy in ES-SCLC has promoted only a modest improvement in OS. Additionally, treatment-induced toxicity is high, often worsening outcomes and decreasing quality of life. In this pharmacogenomic study, we investigated predictive biomarkers of response and toxicity in SCLC treatment. We found that *ERCC1* was significantly associated with PFS and *ERCC2* with thrombocytopenia.

ERCC1 and ERCC2 are rate-limiting enzymes in the NER pathway. Both these enzymes can modulate the therapeutic effects of platinum-based chemotherapy [40]. The predictive value of common genetic variants in their coding genes for efficacy and toxicity has been widely evaluated in lung cancer [23], but research conducted in SCLC is limited despite platinum-based regimens being the standard first-line treatment. Yu et al. [25] identified an intron variant, *ERCC1* rs2298881, associated with response and survival in Asian SCLC patients treated with carboplatin plus etoposide. Nicoś et al. [26] found that patients harboring the AA genotype for *ERCC1* rs11615 had a higher OS than heterozygous Caucasian SCLC patients. Notably, based on ORR and survival, many studies in NSCLC patients have shown significant associations between the *ERCC1* rs11615 variant and the efficacy of chemotherapy [37,41], but others have failed to find a correlation [23].

In the present study we found that LS-SCLC patients carrying the rs11615 G-allele had a longer PFS than those with the wild-type genotype. This finding is consistent with the hypothesis that alterations in the enzymes involved in the DNA repair pathways may lead to better clinical outcomes in patients treated with platinum compounds. Nevertheless, the underlying effect of the variant is poorly understood. The *ERCC1* rs11615 (p.Asn118Asn) polymorphism is a synonymous variant that has been associated with a reduced codon usage and a consequent reduction in ERCC1 mRNA expression [42]. We did not, however, find any correlation between the variant and the expression of the ERCC1 protein or ERCC1-XPF complex, although similar results have already been reported in NSCLC [43]. These finding are in line with previous observations that ERCC1 mRNA and protein levels were not correlated [44,45], thus suggesting the involvement of other factors in the regulation of ERCC1 expression such as post-transcriptional or post-translational modifications. Regardless, *ERCC1* rs11615 may be a feasible biomarker of survival in LS-SCLC.

We also observed that LS-SCLC patients with negative tumoral expression of the ERCC1-XPF complex presented a longer PFS, although the association was not statistically significant probably due to the limited availability of FFPE tissues. Olaussen et al. [18] reported that negative ERCC1 expression, determined by IHC using the 8F1 antibody, was associated with better survival in operated NSCLC patients treated with adjuvant platinum-based chemotherapy. However, subsequent studies failed to confirm this finding [46,47], possibly due to a lack of specificity of the antibody [48]. Moreover, four ERCC1 isoforms, namely 201, 202, 203, and 204, derived from alternative splicing have been described, but only the ERCC1-202 isoform is involved in the repair of platinum-DNA adducts [46]. To deal with this technical challenge, we determined the colocalization in the cell nuclei of ERCC1 and XPF proteins using the PLA technique and the 4F9 antibody. This may be a more reliable method to assess the expression of the endonuclease ERCC1-XPF and overcome the lack of a specific antibody against the ERCC1-202 isoform.

The associations we found between ERCC1, at germline DNA level and at protein expression level, and survival were in LS-SCLC, but not in ES-SCLC. This finding indicates that ERCC1 may be a predictive biomarker of PFS in patients treated with platinum-based chemotherapy plus radiotherapy, supporting findings by other authors [21,49]. ERCC1 may help to identify LS-SCLC patients who could achieve long-term clinical remissions and consequently be potentially curable with standard treatment. ERCC1 may also help to identify patients at high risk of failure of platinum-based treatment. These patients could be selected for inclusion in prospective clinical trials evaluating treatment improvements such as radiotherapy dose escalation [50], or a combination of immunotherapy and chemotherapy. Additionally, they would benefit from maintenance treatments with tarlatamab, lurbinectidin or sacituzumab govitecan that have already demonstrated efficacy in ES-SCLC [3,4,51,52,53,54]. These considerations may be relevant as LS-SCLC patients have limited treatment options, and adding drugs after the completion of platinum-based chemotherapy has not shown to improve outcomes [55].

Another result of interest was the association between the *ERCC2* rs1799793 T-allele and a higher risk of developing grade 3–4 platinum-induced thrombocytopenia in the total cohort. However, this variant was not associated with a dose reduction in chemotherapy or with PFS. As expected, we observed that thrombocytopenia was more frequent in ES-SCLC patients whose regimen of choice was carboplatin–etoposide. Most of these patients required a reduction in the chemotherapy dose and presented poor survival. ERCC2 protein has ATP-dependent 5′-3′ DNA helicase activity and is a subunit of the general transcription and DNA repair factor IIH (TFIIH) core complex. The *ERCC2* rs1799793 variant could identify patients who are at risk of developing thrombocytopenia and would be candidates for preventive treatment with thrombopoietin receptor agonists. To date, this variant has not been previously analyzed as a biomarker of thrombocytopenia in SCLC, but results of the studies conducted in NSCLC were negative [56,57]. The *ERCC2* rs1799793 (p.Asp312Asn) variant results in a change in an amino acid located in the Arch domain (residues 248–438) of the protein. Mutations in this region have been shown to affect DNA repair processes and protein–protein interactions established during transcription [58,59]. Furthermore, Seker et al. [60] observed an increased apoptotic response of lymphoblastoid cell lines homozygous for the Asn312 codon (T-allele) exposed to UV or ionized radiation, supporting *ERCC2* rs1799793 as a putative functional variant.

Our candidate gene-based pharmacogenetic study has several limitations. First, the sample size was relatively small and additionally divided into limited- and extensive-stage SCLC. In addition, the percentage of optimal FFPE tissue available for this study was low because these tissues were prioritized for histological diagnosis and possible clinical trials. Second, most of the informative genetic variants involved in the DNA repair pathways and etoposide metabolism were analyzed in this study, but variants in other relevant genes were not analyzed. The evaluation of additional DNA repair genes such as *ERCC4* or genes encoding for kinases and cyclins involved in the cell cycle, among others, would have allowed us to better describe the interindividual variability in the response to platinum-based chemotherapy and radiotherapy in these patients. Moreover, functional analyses of *ERCC1* rs11615 and *ERCC2* rs1799793 variants remain necessary to better characterize their effect on DNA repair pathways. Lastly, *ERCC1* rs11615 and *ERCC2* rs1799793 associations with survival and toxicity, respectively, did not reach statistical significance according to the Bonferroni correction criteria. Bonferroni correction, however, is a too conservative procedure when correlating genetic markers. To overcome this limitation, we performed complementary ERCC1 protein expression analyses that showed results in line with those observed for the *ERCC1* rs11615 genetic variant. However, the cut-off score for ERCC1-XPF positivity in the PLA technique has yet to be validated.

## 5. Conclusions

Our study adds further evidence to the predictive value of germline variants in genes of the NER pathway in relation to survival and toxicity following platinum-based chemotherapy. ERCC1 is a promising biomarker to personalize platinum-based chemotherapy in LS-SCLC. However, confirmatory findings from larger prospective samples and additional functional analyses of the genomic variants are needed.

## Figures and Tables

**Figure 1 pharmaceutics-16-01121-f001:**
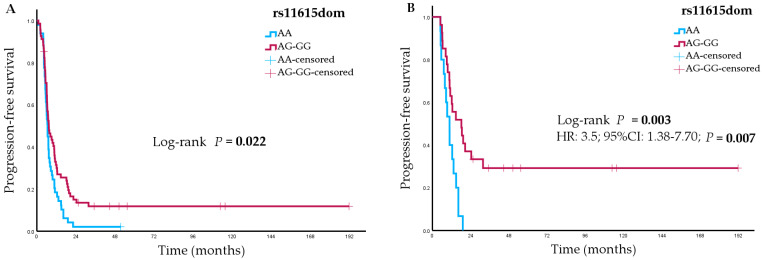
(**A**) Progression-free survival according to *ERCC1* rs11615 variant in the total cohort of small cell lung cancer (SCLC) patients. (**B**) Progression-free survival according to *ERCC1* rs11615 variant in the subgroup of patients with limited-stage disease.

**Figure 2 pharmaceutics-16-01121-f002:**
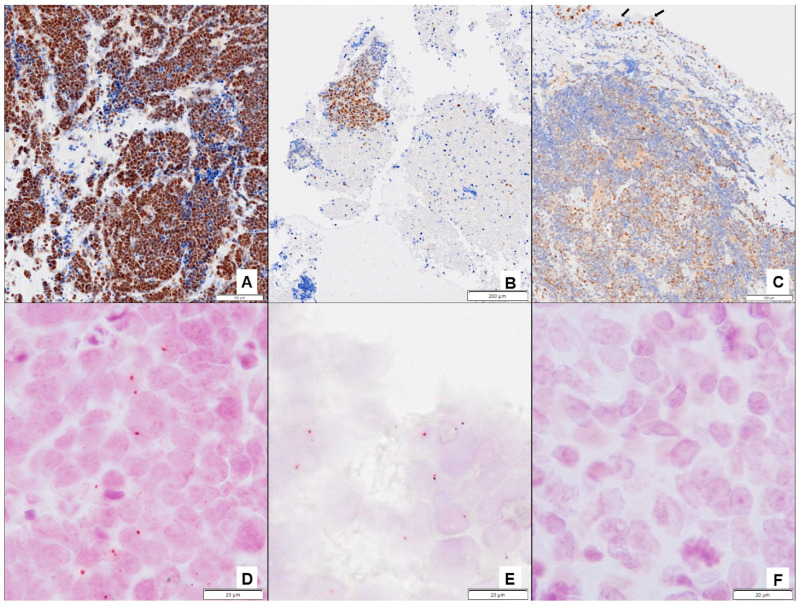
(**A**) High intensity and diffuse ERCC1 nuclear protein expression (clone 4F9, Dako Ltd., Cheshire, UK) by immunohistochemistry (IHC) in a small cell lung cancer (SCLC) biopsy (100×). (**B**) Moderate intensity and diffuse ERCC1 nuclear protein expression (clone 4F9, Dako Ltd., Cheshire, UK) by IHC in a SCLC cell block (40×). (**C**) Low intensity and patched ERCC1 nuclear protein expression (clone 4F9, Dako Ltd., Cheshire, UK) by IHC in a SCLC biopsy. ERCC1 nuclear positivity in bronchial epithelium as internal positive control is marked with a black arrow (100×). (**D**,**E**) Two positive cases with different levels of expression of the ERCC1-XPF complex. The presence of one or two dark red signals locates the ERCC1-XPF complex (ERCC1: clone 4F9, Dako Ltd., Cheshire, UK and XPF: Polyclonal, Novus Biologicals, Colorado, USA) in the SCLC nucleus via proximity ligation assay (PLA) (1000×). (**F**) A negative case for the ERCC1-XPF complex using PLA (1000×).

**Table 1 pharmaceutics-16-01121-t001:** Patients’ baseline clinical characteristics and treatment details.

Characteristic	Total Cohort (n = 145) n (%)	Limited-Stage Disease (n = 45) n (%)	Extensive-Stage Disease (n = 100) n (%)
Baseline characteristics			
Sex			
Male	104 (71.7)	31 (68.9)	73 (73)
Female	41 (28.3)	14 (31.1)	27 (27)
Age, median (range)years	67.4 (38–89)	67 (54–89)	66 (38–86)
ECOG			
0	9 (6.2)	4 (8.9)	5 (5)
1	79 (54.5)	31 (68.9)	48 (48)
2	37 (25.5)	6 (13.3)	31 (31)
3–4	15 (10.4)	1 (2.2)	14 (14)
Missing	5 (3.4)	3 (6.7)	2 (2)
Smoking			
Current	89 (61.4)	22 (48.9)	67 (67)
Former	55 (37.9)	23 (51.1)	32 (32)
Missing	1 (0.7)	0	1 (1)
Metastases			
Brain	20 (14.2)		20 (20)
Liver	43 (29.7)		43 (43)
Treatment details			
Chemotherapy regimen			
Cisplatin	51 (35.2)	22 (48.9)	29 (29)
Carboplatin	94 (64.8)	23 (51.1)	71 (71)
Median cycles (range)	4 (1–6)	4 (1–5)	6 (1–6)
Immune checkpoint inhibitors	9 (6.2%)	0	9 (9.0%)
Radiotherapy			
Thoracic radiotherapy	40 (27.6)	40 (88.9)	
Concomitant	22 (15.2)	22 (48.9)	
Sequential	18 (12.4)	18 (40)	

Eastern Cooperative Oncology Group (ECOG).

**Table 2 pharmaceutics-16-01121-t002:** Description of genetic markers previously associated with clinical outcomes in lung cancer patients treated with platinum compounds plus etoposide.

Gene Symbol	Reference SNP	Variant Description	HGVS (Nucleotide)	MAF Cohort	References for Rationale
*XRCC1*	rs25487	Missense	NM_006297.3:c.1196A>G	0.37	[34]
*ERCC1*	rs3212986	3′-UTR	NM_001369414.1:c.*197=	0.23	[23,26,37]
	rs11615	Synonymous	NM_001369414.1:c.354T>C	0.37	[23,26,37]
*ERCC2*	rs13181	Missense	NM_000400.4:c.2251A>C	0.34	[37]
	rs50872	Intron	NM_000400.4:c.1238–1492T>C	0.26	[36]
	rs1799793	Missense	NM_000400.4:c.934G>A	0.33	[36]
*ABCC3*	rs4793665	2KB upstream	NC_000017.11:g.50634726C>T	0.46	[36]
*ABCB1*	rs1045642	Synonymous	NM_001348944.2:c.3435T>C	0.47	[35,36]
	rs2032582	Missense	NM_001348944.2:c.2677T>G; c.2677T>A	0.41/0.05	[35,36]
	rs1128503	Synonymous	NM_001348944.2:c.1236T>C	0.43	[35,36]
*UGT1A1*	rs3064744	Upstream	NG_002601.2:g.175491_175505	0.36	[27,28]
*GSTP1*	rs1695	Missense	NM_000852.4:c.313A>G	0.34	[37]

SNP: single-nucleotide polymorphisms; HGVS: human genome variation society; MAF: minor allele frequency.

**Table 3 pharmaceutics-16-01121-t003:** Significant univariate associations between genetic variants and survival in the total cohort, limited-stage SCLC and extensive-stage SCLC subgroups.

	Progression-Free Survival				Overall Survival			
SNP	n	mPFS (95%CI), Months	HR (95% CI)	*p*-Value	n	mOS (95%CI), Months	HR (95% CI)	*p*-Value
Total Cohort								
*ERCC1* rs11615								
AA	49	6.6 (5.8–7.4)	Reference (1)	0.06	50	11.3 (7.9–14.7)	Reference (1)	0.26
AG	52	7.0 (4.0–10.0)	0.66 (0.44–1.0)		53	11.8 (6.8–16.8)	0.72 (0.48–1.09)	
GG	16	7.7 (5.8–9.6)	0.57 (0.31–1.03)		18	12.7 (8.9–16.4)	0.75 (0.43–1.31)	
AG-GG ^a^	68	7.4 (5.6–9.1)	0.64 (0.43–0.94)	**0.02**	71	12.7 (8.6–16.7)	0.73 (0.50–1.07)	0.1
*ABCC3* rs4793665								
TT	35	6.4 (5.4–7.4)	Reference (1)	0.18	35	10.1 (7.5–12.7)	Reference (1)	**0.046**
TC	59	7.7 (5.7–9.8)	0.67 (0.43–1.03)		62	14.6 (10.4–18.7)	0.59 (0.38–0.90)	
CC	23	7.3 (6.1–8.6)	0.80 (0.47–1.34)		24	12.7 (7.3–18.0)	0.77 (0.46–1.31)	
Limited-stage disease								
*ERCC1* rs11615								
AA	15	10.8 (8.0–13.6)	Reference (1)	**0.009**	15	20.7 (14.3–27.1)	Reference (1)	**0.04**
AG	17	18.2 (10.2–26.2)	0.3 (0.13–0.68)		18	33.8 (26.8–40.7)	0.38 (0.17–0.83)	
GG	10	9.6 (0.0–28.5)	0.43 (0.17–1.1)		10	14.8 (0.0–45.4)	0.73 (0.32–1.7)	
AG-GG ^b^	27	18.2 (8.1–28.3)	0.34 (0.16–0.71)	**0.003**	28	33.8 (22.2–45.4)	0.48 (0.24–0.96)	**0.03**
*ERCC2* rs50872								
GG	23	13.3 (9.3–17.4)	Reference (1)	**0.03**	24	31.8 (19.4–44.3)	Reference (1)	0.16
GA	14	14.9 (5.5–24.3)	0.66 (0.3–1.45)		14	16.1 (0.0–44.6)	1.00 (0.47–2.13)	
AA	5	7.8 (6.4–9.3)	2.9 (1.03–8.17)		5	15.3 (6.7–23.9)	2.55 (0.92–7.09)	
GG-GA	37	14.8 (10.6–18.9)	0.30 (0.11–0.82)	**0.01**	38	28.7 (16.7–40.7)	0.39 (0.15–1.06)	0.06
*ABCC3* rs4793665								
TT	9	11.2 (9.6–12.8)	Reference (1)	0.24	9	23.0 (0.0–55.8)	Reference (1)	**0.04**
TC	25	14.8 (10.8–18.7)	0.57 (0.26–1.29)		25	35.2 (32.8–37.5)	0.40 (0.17–0.92)	
CC	8	9.6 (4.5–14.7)	1.05 (0.39–2.84)		9	19.0 (7.0–30.9)	0.96 (0.38–2.51)	
*UGT1A1*28* rs3064744								
*1/*1	17	19.1 (15.1–23.1)	Reference (1)	0.05	17	34.8 (29.1–40.6)	Reference (1)	0.23
*1/*28	15	10.8 (6.6–15.1)	2.57 (1.17–5.64)		16	15.3 (7.2–23.3)	1.90 (0.89–4.07)	
*28/*28	10	9.1 (5.7–12.5)	1.84 (0.75–4.54)		10	16.1 (0.0–45.4)	1.17 (0.48–2.86)	
*1/*28–*28/*28 ^a^	25	10.8 (8.0–13.6)	2.24 (1.1–4.57)	**0.02**	26	16.1 (8.8–23.4)	1.559 (0.78–3.10)	0.2
Extensive-stage disease								
*ERCC1* rs3212986								
GG	46	6.0 (4.8–7.2)	Reference (1)	**0.049**	49	10.6 (8.6–12.5)	Reference (1)	0.1
GA	27	5.4 (3.8–6.9)	1.81 (1.08–3.00)		27	7.5 (5.8–9.2)	1.69 (1.04–2.73)	
AA	2	7.7 (NA-NA)	0.67 (0.16–2.75)		2	12.4 (NA–NA)	1.06 (0.26–4.42)	

^a^ Dominant model; ^b^ Recessive model. Statistically significant *p*-values are marked in bold.

**Table 4 pharmaceutics-16-01121-t004:** Significant univariate associations between genetic variants and PE-based chemotherapy-related toxicity in the total cohort.

SNP	Affected, n/ Total, n	(%)	*p*-Value
	Anemia		
*ERCC2* rs50872			
GG	11/74	14.9	**0.04**
GA	9/58	15.5	
AA	4/8	50	
GG-GA ^b^	20/132	15.2	**0.03 ***
*XRCC1* rs25487			
CC	9/52	17.3	0.12
CT	15/71	21.1	
TT	0/17	0	
CC-CT ^b^	24/123	19.5	**0.04 ***
	Thrombocytopenia		
*ERCC2* rs1799793			
CC	6/61	9.8	**0.04**
CT	14/65	21.5	
TT	5/14	35.7	
CT-TT ^a^	19/79	24.1	**0.03**
*UGT1A1* rs3064744			
*1/*1	12/64	18.8	0.05
*1/*28	5/50	10.0	
*28/*28	8/24	33.3	
*1/*1–*1/*28 ^b^	17/114	14.9	**0.03**

^a^ Dominant model; ^b^ Recessive model; * Fisher test. Statistically significant *p*-values are marked in bold.

## Data Availability

The data presented in this study are not publicly available due to ethical committee regulations but are available on request from the corresponding authors.

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
