# Peer review of "ERCC1 and ERCC2 Polymorphisms Predict the Efficacy and Toxicity of Platinum-Based Chemotherapy in Small Cell Lung Cancer"

_pharmaceutics, 2024, doi:10.3390/pharmaceutics16091121_

Round 1

Reviewer 1 Report

Comments and Suggestions for Authors

The manuscript by Barba A. et al. aim to identify predictive biomarkers of  PE-treated SCLC. Although, the main concept of the manuscript is really interesting there are many flaws in this research. 

1) One of the sex main therapeutically targets of  SCLC is cell cycle and DNA damage repair. I would like to know how you choose only these genetic markers. Why did you not include more markers regarding cell cycle?

2) You have included in your study only male SCLC samples. Why? You should also include female samples to prevent gender-specific impact.

3) If you want to prove that ERCC1 and ERCC2 are  predictive biomarkers, you should perform more research experiments. You could carry out a bioinformatic analysis in publicly available TCGA samples and evaluating the effect of ERCC1 and ERCC2 in SCLC samples. Furthermore, you should perform some in vitro experiments (in H69PR cell line or any other preclinical model) creating technically the SNPs polymorphism by site directed mutagenesis and evaluate their effect on DNA repair pathways and or/cell cycle. 

Comments on the Quality of English Language

English language is fine. 

Reviewer 2 Report

Comments and Suggestions for Authors

Small lung cancer is an aggressive type of tumor with poor prognosis. Treatment involves surgery in case of small size tumors or chemotherapy associated with radiation which efficacy depends on genetic variants in DNA-repair genes. Barba et al., in the present study attempted to investigate germline polymorphisms in some of the genes involved in DNA repair, as for example XRCC1, ERCC1, ERCC2, ABCB1, ABCC3, UGT1A1 and GSTP1, in 145 patients diagnosed with small cell lung cancer. ERCC1 expression in tumor tissues was investigated by immunohistochemistry and the proximity ligation assay was employed to study ERCC1-XPF complex. Without a statistical significance, the authors found an association between the ERCC1 rs11615 variant and the median progression-free survival in patients with limited-stage, as well as differences between the ERCC1-XPF complex and the median progression-free survival in limited stage small cell lung carcinoma patients. In addition, a correlation between the ERCC2 rs1799793 variant and thrombocytopenia in the total cohort was observed. The authors concluded that their results provide evidence that ERCC1 and ERCC2 variants may predict efficacy and safety of platinum-based chemotherapy in patients with small cell lung cancer and that patients with limited-stage may benefit by ERCC1 determination.

The manuscript may be clinically relevant but it is not well organized and the results do not fully support the conclusions and must be improved.

The Introduction is poor (e.g., what are ERCC1 and ERCC1? What are their functions? Are any polymorphisms associated with ERCC1/2?).

Allelic frequency distribution analysis and discrimination are missing, baseline characteristics of the patients (e.g., blood parameters, liver and kidney functions…) before and after treatment must be included, the effects of ERCC1 rs11615 polymorphism on post-chemotherapy blood levels in determining treatment outcomes for small cell lung cancer patients should be reported. I would suggest the authors to also include a Table with the selected SNPs genes and the prognostic effects of the variants in small cell lung cancer. Table 2 in the manuscript is not complete and the variants are not linked to any prognostic effect. The proximity ligation assay is not convincing and should be properly analyzed. Why do the study include only male patients? Please, explain it. All Figures should be adequately labeled and presented (e.g., in Figure 1 A the genetic variants have different evaluation time, scale bars in Figure 2 are missing, as well as the ICH staining markers…etc). Material and Methods are not adequately described (e.g., primary and secondary antibodies are missing, tissue information and preparation are lacking... etc).

Strengths and weaknesses of the research must be better discussed.

Comments on the Quality of English Language

English is fine.

Reviewer 3 Report

Comments and Suggestions for Authors

Dear Authors, a very interesting study on ERCC1/ERCC2 polymorphisms in small cell lung cancer. The manuscript is prepared with attention to detail, and the content is very transparent (I appreciate a comprehensive description of statistical analysis, and presenting significant and non-significant data), which results in minor revisions on my side. Thus, please answer or consider the following:

(1) Introduction, line 42: Add “of” between “half” and “the patients”.

(2) Introduction, line 51: Consider changing “with higher toxicity and poorer survival [6,7], and comorbidities and high pretreatment” to “with higher toxicity and poorer survival [6,7], as well as comorbidities and high pretreatment” (there are too many “and” next to each other).

(3) Introduction, line 61: “In vitro” should be italicized. Taking this chance, please double-check the entire manuscript and look for similar examples, non-italicized gene symbols, etc. But in general, the manuscript is well prepared in this context.

(4) Why XPF polymorphisms were not investigated in this study? I assume there is no literature data to associate it with clinical outcomes in lung cancer patients treated with PE-based chemotherapy and/or radiotherapy (which was the case for selected SNPs as mentioned in section 2.3).

(5) Out of curiosity – do you have any matched normal tissue specimens acquired from your 145 patients? If yes, what kind of research do you plan to perform in the future?

(6) Materials and Methods, Table 1: “ECOG” is used for the first time but this abbreviation is explained later in the text (in line 177). Please explain “ECOG” in the footnote of Table 1 or somewhere close.

(7) Abbreviations and their explanation on first use should be improved throughout the entire manuscript. “RECIST” should be explained in line 99. I would also write “interquartile range” instead of “IQR” in line 190 because it is not used elsewhere in the text. “ORR” could be explained earlier in the text (Abstract) than in line 101, although this comment is optional. “IHC” could be explained earlier in the text (Abstract) than in line 133, although this comment is also optional.

(8) Results, line 119: Consider adding a column in Table 2 to include references. I would also make a title more detailed (something like “Genetic markers previously associated with clinical outcomes in PE-treated lung cancer patients”).

(9) Results, lines 215-243: Try to decrease the size of survival curves in Figure 1 so the entire graph would fit on a single page together with the figure’s description. Enlarge the font to avoid making the text illegible. (there is plenty of free space above survival curves).

(10) Results, line 278: Add a scale to images in Figure 2.

(11) Discussion, lines 350-351: Consider changing the text to “Given that the associations between survival and ERCC1, either at germline DNA level or at the protein expression level, were found in LS-SCLC and not in ES-SCLC”.

(12) Discussion, lines 387-388: Change “And fourth” to something like “Lastly” or “Ultimately” at the beginning of this sentence.

Comments on the Quality of English Language

Minor English revisions that I mentioned in my specific comments.

Round 2

Reviewer 1 Report

Comments and Suggestions for Authors

Dear Authors,

Thank you for taking my comments into account. I have read all your responses and understand your perspective. However, I must insist that additional experiments are needed to enhance your results and improve the overall quality of your manuscript. Nonetheless, I suggest publishing your interesting paper as a Communication rather than as an Article.

Reviewer 2 Report

Comments and Suggestions for Authors

The authors addressed almost all the concerns raised, and the manuscript has improved.